# MicroRNA Profiling Shows a Time-Dependent Regulation within the First 2 Months Post-Birth and after Mild Neonatal Hypoxia in the Hippocampus from Mice

**DOI:** 10.3390/biomedicines10112740

**Published:** 2022-10-28

**Authors:** Aisling Leavy, Gary P. Brennan, Eva M. Jimenez-Mateos

**Affiliations:** 1Discipline of Physiology, School of Medicine, Trinity College Dublin, The University of Dublin, D02 R590 Dublin, Ireland; 2Conway Institute, School of Biomolecular and Biomedical Science, University College Dublin, Belfield, D04 C7X2 Dublin, Ireland

**Keywords:** microRNAs1, hippocampus, hypoxia

## Abstract

Brain development occurs until adulthood, with time-sensitive processes happening during embryo development, childhood, and puberty. During early life and childhood, dynamic changes in the brain are critical for physiological brain maturation, and these changes are tightly regulated by the expression of specific regulatory genetic elements. Early life insults, such as hypoxia, can alter the course of brain maturation, resulting in lifelong neurodevelopmental conditions. MicroRNAs are small non-coding RNAs, which regulate and coordinate gene expression. It is estimated that one single microRNA can regulate the expression of hundreds of protein-coding genes.. Uncovering the miRNome and microRNA-regulated transcriptomes may help to understand the patterns of genes regulating brain maturation, and their contribution to neurodevelopmental pathologies following hypoxia at Postnatal day 7. Here, using a PCR-based platform, we analyzed the microRNA profile postnatally in the hippocampus of control mice at postnatal day 8, 14, and 42 and after hypoxia at postnatal day 7, to elucidate the set of microRNAs which may be key for postnatal hippocampus maturation. We observed that microRNAs can be divided in four groups based on their temporal expression. Further after an early life insult, hypoxia at P7, 15 microRNAs showed a misregulation over time, including Let7a. We speculated that the transcriptional regulator c-myc is a contributor to this process. In conclusion, here, we observed that microRNAs are regulated postnatally in the hippocampus and alteration of their expression after hypoxia at birth may be regulated by the transcriptional regulator c-myc.

## 1. Introduction

Human brain development begins in the 3rd week of gestation and continuing until adulthood [1], with dynamic changes in the brain happening through childhood and adolescence. The complex process that is postnatal brain maturation involves the formation and maturation of tracts, combined with regressive processes including apoptosis of brain cells [2,3]. Whilst the generation and migration of neurons is mainly a prenatal event, the neuron-glia interactions and the functional organization of neural circuits happens during postnatal life [4]. In particular, in humans during school-age years (4–11 years old), maturation of the brain’s connectivity tract is prominent [5]. In the postnatal period, neurogenesis is limited and focused within specific areas, such as the olfactory bulb and the dentate gyrus of the hippocampus [6]. Gliogenesis, on the other hand, is very prominent during the postnatal period. Glial progenitors proliferate in the forebrain subventricular zone (SVZ) and migrate to the cortex, striatum, and hippocampus. Here, they differentiate into oligodendrocytes and astrocytes, forming the myelin sheaths and providing homeostatic support to neurons and enabling the maturation of neuronal circuits [7,8]. Furthermore, synaptic pruning and production of neuronal connections happen prominently during the first years of life. In fact, across the whole brain, the number of synapses is double in childhood than in adulthood, demonstrating that the processes of synaptic remodelling are very active during childhood and adolescence, and these processes involve the generation and, removal of synapses [4]. Despite the current understanding of the overall process, the molecular mechanisms dictating postnatal brain are not well understood.

MicroRNAs are small non-coding RNAs (22–24 nucleotides) that regulate gene expression post-transcriptionally, by inhibiting translation or promoting mRNA degradation through 3′UTR binding of the target mRNA [9]. One single microRNA can regulate the translation of hundreds of protein-coding mRNAs. Although, bioinformatics predictor models show that most of the protein-coding genes are regulated by combinations of microRNAs, rather than one single microRNA, therefore, unravelling microRNA transcriptomes are necessary to understand gene regulation of complex processes [9]. Importantly, microRNAs can be grouped in families, these microRNAs share target genes, and have reductant functions [10].

MicroRNAs are essential for normal development, using genetic techniques it was observed that disruption of microRNA production results in embryonic lethality in zebrafish, worms, flies, and mice, mainly due to neuronal deficits [11,12,13,14]. MicroRNAs are widely implicated in the regulation and homeostasis of early brain development, including embryonic neurogenesis and neuronal differentiation. The complexity of brain maturation, with several processes happening over years in humans, suggests that families of microRNAs may be involved in directing brain development and maturation [15,16], however only few studies have analysed the role of microRNAs in specific areas of the brain postnatally, e.g., hypothalamus [17]. Nevertheless, the specific function of a single microRNA may not be critical, as these microRNAs may have very high temporal specificity and cellular expression, and their function may be compensated by other cells in complex multicellular organisms [18]. To fully understand the normal development of complex organs, such the brain, it is important to analyse the expression of microRNAs at several time points.

C-myc is a highly conserved transcription factor belonging to the MYC family of basic helix-loop-helix transcription factors and controls the transcriptional networks that govern cell growth, division, differentiation, and death [19]. MYC regulates extensive transcriptional programs and drives cells to pre-determined cellular states by amplifying cell-specific transcriptional programs [19]. MYC mRNA has been identified in all multiple neural cell types including, neurons, radial glia cells, and oligodendrocytes, suggesting it has a key function in the formation of brain circuits. Transcriptional control by MYC is complex, it can promote and repress transcription depending on the cellular context [19]. In this aspect, MYC protein inhibits the transcription of several microRNA clusters [19], demonstrating that activation of MYC will result in the amplification of signaling pathways. In fact, in human hepatocellular carcinoma cells, MYC represses the expression of the let-7a cluster by binding to the non-canonical E-box [20] and, in the brain, MYC represses the expression of the miR-23 cluster, which may result in a reduction in myelination [21].

In this study, we aim to evaluate how microRNA expression changes over time in the hippocampus and how an early life insult, hypoxia at Postnatal day 7 may result in the misregulation of microRNAs and contribute to the neurological outcomes. To this aim we analyzed the microRNA profile in the hippocampus from Postnatal day 8 (P8) to Postnatal day 42 (P42) in mice, and how microRNA expression changed after perinatal stress using a mouse model of hypoxia at Postnatal day 7 (P7). In control, we observed that microRNAs can be divided into 4 different groups, microRNAs with stable expression during aging, microRNAs with either increased or decreased expression between ages, and microRNAs with a peak in the middle time point (P14) compared to P8 and P42. Additionally, we observed that after hypoxia at Postnatal day 7, a sub-set of microRNAs changed the expression over time, including the let-7a cluster. Here, we speculated that while these happen at specific time points, the overall summation and accumulation may contribute to the pathology after hypoxia in the neonatal period. Finally, bioinformatics tools were validated using chromatin immunoprecipitation (ChiP) assay, and we identified that the transcriptional controller c-myc may regulate the expression of two of these microRNA clusters, let-7a and miR-23.

## 2. Materials and Methods

### 2.1. Mouse Model of Neonatal Hypoxia

A total of 70 mice were used in this study. All animal procedures were performed in accordance with the principles of European Communities Council Directive (86/609/EEC, 2010/63/EU), under license (REC#1203b or REC#P140) from the Department of Health and Health Products Regulatory Authority (Ireland) and procedures were approved by the Research Ethics Committee of the Royal College of Surgeons in Ireland and Trinity College Dublin. Neonatal litters of C57BL/6J mice [weight, 4–6 g; age, postnatal day 6.5–7.5 (P7)], were obtained from the Biomedical Research Facility, RCSI or CMU at TCD. Pups were kept with their dams in a barrier-controlled facility on a 12 h light–dark (7 a.m.–7 p.m.) standard cycle with access to food and water ad libitum. All experiments were performed during the light cycle.

Hypoxia in mouse pups was carried out as previously described [22]. Male and female pups from the same litter were randomly placed in a clear hypoxic chamber and exposed to a premixed gas containing 5% O2/95% N2 for 15 min at 34 °C and 80% humidity. Under these conditions, 97% of the pups develop behavioural seizures during and after hypoxia. All animals were observed during the 15 min of hypoxia and 15 min post-hypoxia before returning to the dam. Both sexes were used for consecutive studies.

### 2.2. RNA Isolation and Open Array

For each time point and experimental condition, mice were perfused with cold PBS and both hippocampi were isolated and pooled together for RNA extraction. Total RNA was extracted using the Trizol method [22,23,24]. The quality and quantity of RNA were measured using a Nanodrop Spectrophotometer (Thermo Scientific, Waltham, MA, USA) and samples with an absorbance ratio at 260/280 between 1.8–2.2 were considered acceptable.

500 ng of RNA was processed by reverse transcriptase and pre-amplification steps following the manufacturer’s protocol (Applied Biosystems, Waltham, MA, USA). The pre-amplification reaction was mixed with TaqMan OpenArray Real-Time PCR Master mix (1:1). The mix was loaded onto the OpenArray custom-designed panel (215 mature highly expressed in brain miRNAs) and ran using a QuantStudio 12 K Flex PCR (Life Technologies, Carlsbad, CA, USA). The number of miRNA identified in each sample showed good consistency: 136, 132, and 146 in control samples (P8, P14, and P42 respectively), and 153, 131, and 147 in hypoxia samples (P8, P14, and P42 respectively). Any microRNA with a Ct value above 28 was considered “not detected” and it must be expressed in 3 out of 4 samples to be considered detected within a group. After this selection criteria, 157 microRNAs were detected from the original 215 miRNA. The mean of the Ct values was (i) in control samples, P8: 20.87, P14: 20.15, and P42: 20.92 and (ii) in hypoxic samples, P8: 20.64, P14: 20.35, and P42: 19.29 (Figure 1A).

The Geometric Mean Normalisation Method (GMN) and 2^−ΔΔCT^ was used to determine the relative quantification of each target miRNA expression, using for each microRNA the average of control samples at P8 as a reference to calculate the ΔΔCT.

miRNAs with a fold change value of under 0.6 or over 1.5 were deemed significant. Heatmaps were compiled to represent the fold change values for each miRNA sample group over the three time points, allowing miRNAs to be grouped according to their trends in expression over time (Figure 2).

### 2.3. Ambulation Score

Pups were placed in a flat, enclosed area and recorded for three minutes. Immobile pups were encouraged to move by gently prodding the back of the pup. The ambulation scoring scale was adapted from Feather-Schussler et al. [25]: 0 = no movement, 0.5 = lots of turning, encouragement needed, little or no straight crawling, 1 = crawling with asymmetric limb movement, 1.5 = crawling, mostly asymmetric with sporadically symmetric limb movement, 2 = slow crawling with symmetric limb movement, 2.5 = fast crawling with symmetric limb movement, and 3 = walking with symmetric limb movement. Symmetric limb movement was noted when the pups’ hind paws met the front paws when moving in a fluid and continuous motion. Asymmetric limb movement was described as erratic placement of front and hind paws when moving, in addition to a lack of fluidity. Crawling was differentiated from walking based on what portion of the hindlimb was in contact with the ground when ambulating. During crawling, the entire hind paw and heel make contact with the ground. While only the toes come into contact with the ground. The maximum score during the 3 min was given [25].

### 2.4. RT-PCR of Primary Let-7a Cluster Sequence

250 ng of total RNA was retro transcribed using the High-Capacity cDNA Reverse Transcription kit (Thermo Fisher, Cat n: 4368814) following the manufacturer’s protocol. Two microlitres of RT product were amplified using TaqMan Fast Universal Master Mix 2x (Thermo Fisher, Cat n: 4352042) following manufacturer instructions. Pre-designed primers for the primary sequence and the housekeeping gene (actin) were purchased from Thermo Fisher (Let7a primers: Mm03306744pri and Actin: Mm00306184pri). Let7a levels were normalized to actin (housekeeping gene) and data was represented as a relative expression to the control group using the 2^−ΔΔCt^ method [23].

### 2.5. Identification of c-Myc Binding Sites in the Promoter Regions

The promoter region from the 15 misregulated microRNAs over time in the hypoxia group compared to the control was examined for potential Myc binding sites. Potential C-Myc binding sites were identified by searching for Myc-binding motifs within 5000 bases upstream of the transcriptional start sites of genes of interest, using the UCSC genome browser (version: GRCm39/mm39). Primers were then designed to flank the predicted Myc binding sites.

### 2.6. Chromatin Immunoprecipitation (ChIP)

ChIP was performed as previously described [23]. Briefly, both hippocampi were extracted 72 h after hypoxia or normoxia conditions. Both hippocampi were pooled and homogenized in 1% formaldehyde (Thermo Fisher Scientific, Rockford, IL, USA) and incubated at room temperature for 10 min. Formaldehyde was quenched with 0.125 M glycine (Sigma-Aldrich Ltd., Wicklow, Ireland). Samples were then incubated in a hypotonic buffer (20 mM Tris-HCl, 10 mM NaCl, 3 mM MgCl2), lysed in a dounce homogenizer and rotated at 4 °C for 20 min. 1% NP-40 (MP) was added to each sample, then samples were centrifuged at 15,000 rpm for 5 min at 4 °C and chromatin was sheared in a Diagenode Pico sonicator. To ensure appropriate shearing profiles, DNA was analyzed on a 2% agarose gel and visualized on a UV box (VWR International Ltd., Dublin, Ireland). Then, samples were incubated with magnetic Dynabeads^®^ (Thermo Fisher Scientific, Rockford, IL, USA) which had been pre-incubated overnight with 5 μg of c-myc antibody (Merck Millipore, Billerica, MA, USA) or IgG control (Cell Signaling, Danvers, MA, USA). Beads and DNA were incubated at 4 °C for 1 h. Magnetic beads and bound material was washed with RIPA buffer to remove loosely bound DNA/proteins. Beads were then separated from complexed DNA using Chelex reagent (Sigma-Aldrich Ltd., Wicklow, Ireland) and heated to 100 °C for 10 min. Samples were then incubated at room temperature for a further 10 min before being centrifuged at 15,000 rpm for 5 min. The supernatant was transferred to a new tube, stored at 4 °C and quantified by qPCR with gene-specific primers. Transcription factor occupancy in control and post-hypoxia were normalized to IgG binding to the DNA and calculated as a percentage of total input.

Primer sequence: miR23b-27 site 1 (F: 5′GCAGTGGGTGCCTGTAAG3′, R: 5′CTGTTTGGCTCCGTTTCG3′); miR23b-27 site 2 (F: 5′GGGACTTGATGAAATGAGC3′, R: 5′TGAACTTAACTGTGGGTTG3′); Let-7a-1 site 1 (F: 5′TCTCAGGGCCAGAACACTTG3′, R: 5′TGGTGCTAAAAGGCAACCCA3′); Let-7a-1 site 2 (F: 5′TGTCTCATGAACATCTTTTCCTACT3′, R: 5′GCTGGTGTGTGATATCGAGC3′).

### 2.7. Statistical Analysis

All data are presented as box and whiskers. Two group comparisons were made using an unpaired Student’s two-tailed *t*-test (GraphPad Prism). Multiple group comparison was made using One-way Anova, and Bonferroni post hoc test. Significance was accepted at *p* < 0.05.

## 3. Results

### 3.1. MicroRNA Expression Profiles in the Postnatal Hippocampus

Postnatal brain development is critical for the formation and maturation of neuronal circuits. Whilst the overall process of postnatal brain maturation is understood, little is known about the molecular mechanisms guiding postnatal brain maturation. Here, we examined the regulation of 215 microRNAs on postnatal days 8, 14, and 42 to evaluate their expression over the first weeks of life. 155 microRNAs were expressed within at least one condition (Figure 1A), and from those 130 microRNAs were detected in all samples (Figure 1B). One microRNA was only detected at P8 (miR-10b-5p (Figure 1B)), 5 microRNAs were only detected at P14 (e.g., miR-191-3p, miR-216a-5p (Figure 1B)), and one microRNA was only detected at P42 (miR-770-5p (Figure 1B)). Furthermore, 5 microRNAs were detected at the earlier time points (P8 and P14, (Figure 1B)) and, 13 microRNAs were detected at the later timepoints, P14 and P42 (Figure 1B). No microRNAs were detected in both P8 and P42 exclusively.

**Figure 1 biomedicines-10-02740-f001:**
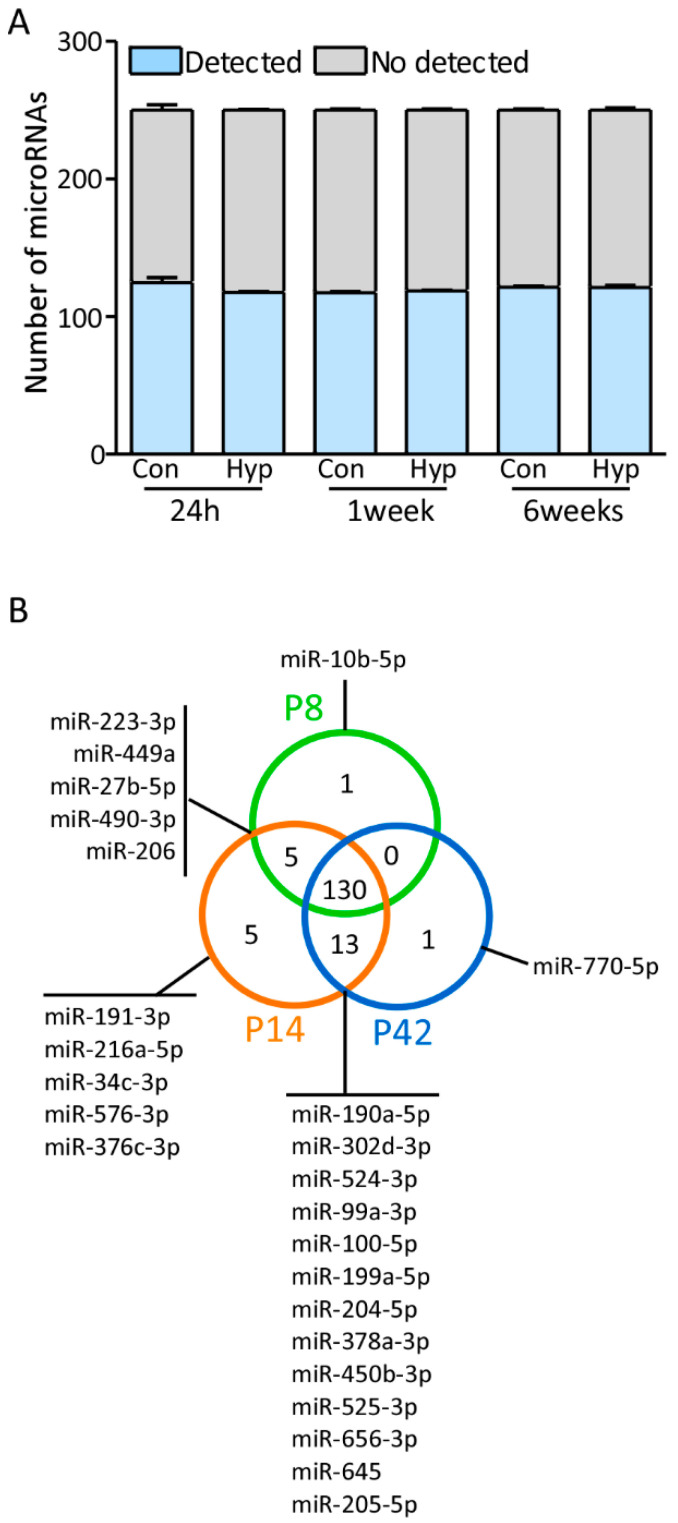
MicroRNA profile in the hippocampus and comparison between ages. (**A**) Graphs show the average number of detected microRNAs (blue) and non-detected (grey), in each experimental group. Con: Control; Hyp: Hypoxia; 24 h: 24 h post-hypoxia (Postnatal day 8, *n* = 4); 1 week: 1 week post-hypoxia (Postnatal day 14, *n* = 4); 6 weeks: 6 weeks post-hypoxia (Postnatal day 42, *n* = 4). (**B**) Ven diagram shows the average of number of detected microRNAs at P8 (green), P14 (orange) and P42 (blue) in the control group.

### 3.2. MicroRNAs Regulation between Ages in the Postnatal Hippocampus in Mice

Then, we analyzed if we could observe trends in the expression of microRNAs between ages (Figure 2). Detected microRNAs were divided into four distinct groups: Group I, microRNAs with stable expression across all ages (Figure 2A); Group II, microRNAs with increased expression at P14 and P42 compared to P8 (Figure 2B); Group III, microRNAs with a peak in expression at P14, with higher or lower expression at P14 compared to P8 and P42 (Figure 2C); and Group IV, microRNAs with peak expression at P8, i.e., lower expression in the latest time point compared to P8 (Figure 2D). Among the 155 microRNAs expressed in all samples, 43 microRNAs were in Group I, and had stable microRNA expression between ages (Figure 2A) including miR-142-3p and miR-148a-3p (Figure 2E). 43 microRNAs were included in Group II with a higher expression in the later time points compared with the earlier ones (Figure 2B), for example, miR-150-5p and miR-219a-5p (Figure 2F); 25 microRNAs had a peak at P14 (higher or lower expression than in P8 and P42) (Group III, Figure 2D), including miR-323a-3p and miR-369-3p (Figure 2G). Finally, in Group IV, 27 microRNAs had a decreased expression over time (Figure 2D), such as miR-106b-5p and miR-19a-3p (Figure 2H).

**Figure 2 biomedicines-10-02740-f002:**
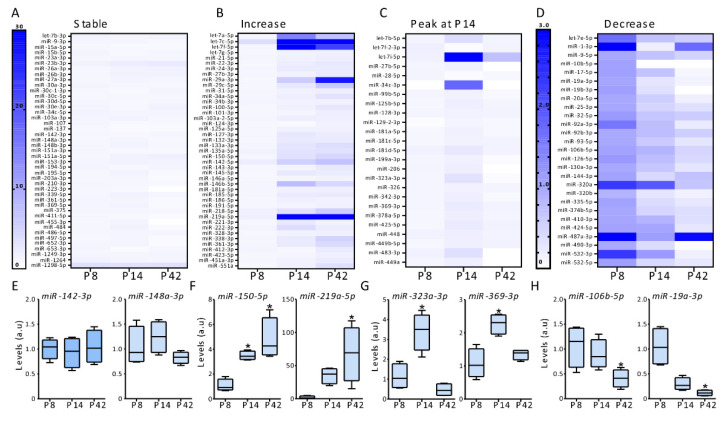
Expression of microRNAs between ages. (**A**) Heat map shows relative expression levels of microRNAs with stable expression at postnatal age 8, 14 and 42 (*n* = 4/group). (**B**) Heat map shows relative expression of microRNAs with increase expression between ages (*n* = 4/group). (**C**) Heat map shows relative expression of microRNAs with a peak in expression at P14 compared to P8 and P42 (*n* = 4/group). (**D**) Heat map shows relative expression of microRNAs with decrease expression between ages (*n* = 4/group). (**E**) Graphs show relative expression of two microRNAs with stable expression between ages miR-142-3p (left) and miR-148a-3p (right) (*n* = 4/group). (**F**) Graphs show the relative expression miR-150-5p (left) and miR-219a-5p (right) which shows an increase expression between ages (*n* = 4/group). (**G**) Graphs show the relative expression of miR-323a-3p (left) and miR-369-3p (right) at P8, P14 and P42 (*n* = 4/group). (**H**) Graphs show the relative expression of two microRNA with decrease expression between ages, miR-106b-5p (left) and miR-19a-3p (right) (*n* = 4/group). One-way ANOVA, * *p* < 0.05.

### 3.3. Differentially Regulated microRNAs after Hypoxia at P7 in Mice

Subsequently, we evaluated if early life stress may disrupt normal developmental miRNA expression profiles. Previously we have shown that pups under hypoxia develop long-term neurological outcomes, including, anxiety-like behavior, hippocampal dysfunction, and a lower threshold to develop seizures [22]. Corroborating these results, P7 pups were subjected to hypoxia (Figure 3A),ambulation test was performed 72 h post-procedure to evaluate neurodevelopment, pups post hypoxia received lower ambulation scores than the control pups (Figure 3B). This demonstrates that hypoxia induces a delay on neurodevelopment. In a separate set of mice, P7 pups were subjected to hypoxia, and hippocampi were dissected 24 h, 7 days, and 5 weeks post-hypoxia corresponding to P8, P14 and P42, respectively, in the control group (Figure 3A). 32 microRNAs were differentially regulated at any given time point. Not surprisingly, the greatest changes in expression were observed 24 h after hypoxia; 22 microRNAs were up-regulated, and no microRNAs were observed to be down-regulated (Figure 3C). Among these microRNAs, we observed an increase in the levels of miR-146b-3p, an inflammation regulated microRNA (Figure 3D), and let-7a-5p, a developmentally regulated microRNA (Figure 3E). 7 microRNAs were differentially expressed 1-week post-hypoxia (Figure 3F); 3 microRNAs were up-regulated, and 4 microRNAs were down-regulated, including miR-29c-5p and miR-532-5p (Figure 3G,H). Finally, 3 microRNAs were dysregulated 5 weeks post-hypoxia (Figure 3I); 2 microRNAs were up-regulated, let-7f-5p (Figure 3K) and miR-124-3p (Figure 3K), and one microRNA was down-regulated. Further analysis showed that the majority of these microRNAs showed a similar trend in expression over time in the control group, such as miR-532-5p (Figure 3H). However, a subset of microRNAs showed a different trend in expression overtime post- hypoxia compared to the control group, even if they were dysregulated in one single time point.

Following this, we decided to analyze if microRNA showed an overall different trend in regulation through the ages post-hypoxia compared to the control group (Figure 4A). We observed that 15 microRNAs, including let-7a (Figure 3E), let-7f (Figure 3J), let-7g, and miR-222 have increased levels over time or showed a peak at P14 in the control group, however, after hypoxia, these microRNAs had a stable expression or decreased expression over time (Figure 4B,C).

### 3.4. MYC Has a High Affinity for Let-7a Promoter in the Hippocampus after Hypoxia

To evaluate if these 15 microRNAs have a common regulator, bioinformatics evaluated common transcriptional sequences at the promoter sites. Bioinformatics analysis showed that these 15 microRNAs had canonical and non-canonical binding sites for c-myc within the promoter areas (Figure 4B). ChIP assays against c-myc were performed in hippocampus 72 h post-procedure (hypoxia or control) and the affinity to let-7a and miR-23 was analyzed. As previously seen in other cell types, c-myc binding was increased post-hypoxia on the first non-canonical Ebox within the Let7a cluster (Figure 4C), and similar results were found when the miR-23a cluster was evaluated (Figure 4D). To further demonstrate that c-myc acts as a transcriptional repressor, we evaluated the primary sequence of let7a, corroborating c-myc affinity results, let7a primary sequence was downregulated in the hypoxia group compared to the control group 72 h post-hypoxia (Figure 4E).

## 4. Discussion

In the current study, we created a neurodevelopmental microRNA profile of 215 brain-expressed microRNAs in hippocampus at 3 different ages, postnatal days 8, 14 and 42. We showed that microRNAs cluster according to expression over time into 4 different groups, stable, up-regulated, down-regulated and showing a peak at the middle time point. Further, when pups were subjected to hypoxia at Postnatal day 7, the regulation of microRNAs was altered, with 15 of these microRNAs regulated by the transcription factors c-myc. 

Corroborating these results, previous studies analyzed the role of microRNAs on the development of the layers of the medial entorhinal cortex postnatally, and age-specific microRNA regulation was identified in rats [26]. MiR-20a, miR-29c, miR-132 and miR-219-5p in particular, were found to be up-regulated in the medial entorhinal cortex, suggesting that these microRNAs may be crucial for neuronal and glial maturation postnatally, including neuronal maturation, dendritic spines morphology and inhibit apoptosis in neurons [27,28], interestengly miR-219 has a key role on myelin formation by regulating oligodendrocytes maturation [29]. This suggests that these microRNAs may be key to the maturation and establishment of neuronal circuits independent of the brain region and species.

Whilst, neurogenesis is more prominent during prenatal development, glia-genesis mainly occurs during postnatal development. In our study and previously, miR-20a was observed to be downregulated over time [26]. MiR-20a is critical for brain maturation during embryogenesis by regulating axonal growth and dendritogenesis, and its expression is time-dependent [30]. Current results support the idea that miR-20a is controlled during early brain development and its expression is downregulated postnatally [30]. Let-7a and miR-125, are microRNAs involved in astrocyte and microglia homeostasis and were observed to be up-regulated between ages in our study, strengthening the hypothesis that glia maturation is more prominent postnatally.

Postnatal brain development is a time-dependent process, genes are expressed in very specific patterns, and the accumulation of malfunctioning genes may cause neurological conditions. After neonatal hypoxia, we observed up-regulation or downregulation of microRNAs at one specific time, but more importantly, we determined patterns of expression with microRNAs being up- or down-regulated over time. We suggest that these cumulative effects may be more detrimental over time than at a single time point. Further, c-myc is a master regulator of gene expression, and through c-myc, the detrimental signal may be amplified and alter several pathways in parallel, thus reducing the possibility of compensatory signals to maintain homeostasis. Finally, c-myc is a transcription factor that previously has been involved in hypoxic responses [31]. Whilst this work has been mainly done in cancer, we speculate that will also be implicated in the brain.

The current study presents some limitations, first, we analysed a small number of microRNAs, selected by their expression on the postanal brain, we cannot discard that important microRNAs involved mainly in the maintenance of brain function, may have a higher expression at P42, and were not evaluated in the current study. Future studies should evaluate microRNA expression in elder mice to consider that microRNA expression may also be regulated from adulthood to the elderly. In the current study we analyzed male and female mice together, we observed no differences in expression between mice, however we cannot discard sex differences. Further studies should evaluate sex differences during postnatal brain development to evaluate the effect of sex in microRNA expression. Finally, we focus on the hippocampus, a vulnerable region to hypoxia, similar changes in specific microRNAs have been seen in other brain regions, e.g., entorhinal cortex [26], we can speculate that common mechanisms are activated independent of the region, however, it will be necessary to evaluate their regulation in parallel to determine if the time scale of brain maturation is similar.

## 5. Conclusions

Postnatal brain development is a sensitive time regulated process, and while acute dysregulation of a single microRNA may be compensated by other pathways, the accumulation effects of increasing dysregulation between ages could have deleterious consequences, particularly, when microRNAs can regulate thousands of pathways and may amplify the detrimental signals. In this study, we showed how microRNAs are expressed across different ages postnatally in the hippocampus, which may be critical for understanding the underlying mechanisms behind the correct brain maturation and neuronal circuits. We also observed that 15 microRNAs have an altered expression over time following hypoxia possibly due to the activity of c-myc, and these changes may result in long-lasting changes during brain maturation.

## Figures and Tables

**Figure 3 biomedicines-10-02740-f003:**
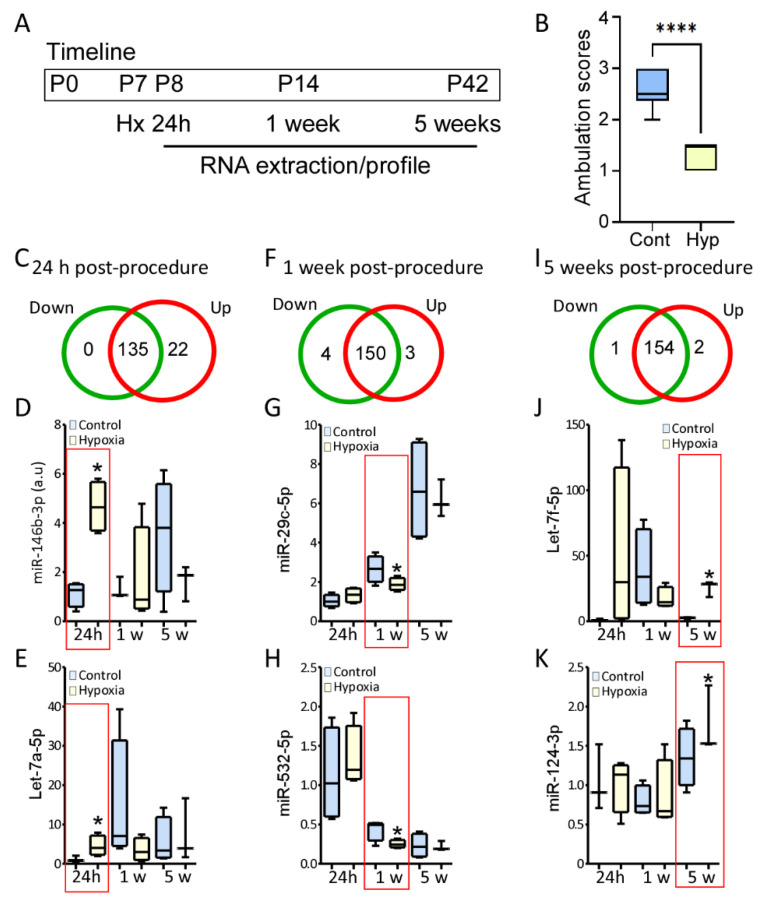
MicroRNA expression post-hypoxia compared to control at different age. (**A**) Schematic showing the timeline of the experimental procedures. (**B**) Pups subjected to hypoxia have a lower ambulation score 72 h post-hypoxia compared to the control pups, demonstrating that hypoxia results in a neurodevelopmental delay. (**C**) Venn-Diagram shows the number of down-regulated (left green), no-regulated (middle, green and red area) and up-regulated (right red) 24 h post-procedure. (**D**,**E**) Graphs show relative expression of miR-146b-3p (**D**) and let-7a-5p (**E**), two of the microRNAs up-regulated 24 h post procedure (*n* = 4/control group; *n* = 4, 24 h and 1 week post-hypoxia; *n* = 3, 5 weeks post-hypoxia. (**F**) Venn-Diagram shows the number of down-regulated (left green), no-regulated (middle, green and red area) and up-regulated (right red) 1 week post-procedure. (**G**,**H**) Graphs show relative expression of miR-29c-5p (*n* = 3–4/group) (**G**) and miR-532-5p (*n* = 4/control group; *n* = 4, 24 h and 1 week post-hypoxia; *n* = 3, 5 weeks post-hypoxia) (**H**), two microRNAs with lower expression 1 week post procedure. (**I**) Venn-Diagram shows the number of down-regulated (left green), no-regulated (middle, green and red area) and up-regulated (right red) 6 week post-procedure. (**J**,**K**) Graphs show relative expression of let-7f-5p (*n* = 4/control group; *n* = 4, 24 h and 1 week post-hypoxia; *n* = 3, 5 weeks post-hypoxia) (**J**) and miR-124-3p (*n* = 4/control group; *n* = 4, 24 h and 1 week post-hypoxia; *n* = 3, 5 weeks post-hypoxia) (**K**), the two microRNAs up-regulated 5 week post-procedure. (*n* = 4/control group; *n* = 4, 24 h and 1 week post-hypoxia; *n* = 3, 5 weeks post-hypoxia). One-way Anova, * *p* < 0.05. **** *p* < 0.001.

**Figure 4 biomedicines-10-02740-f004:**
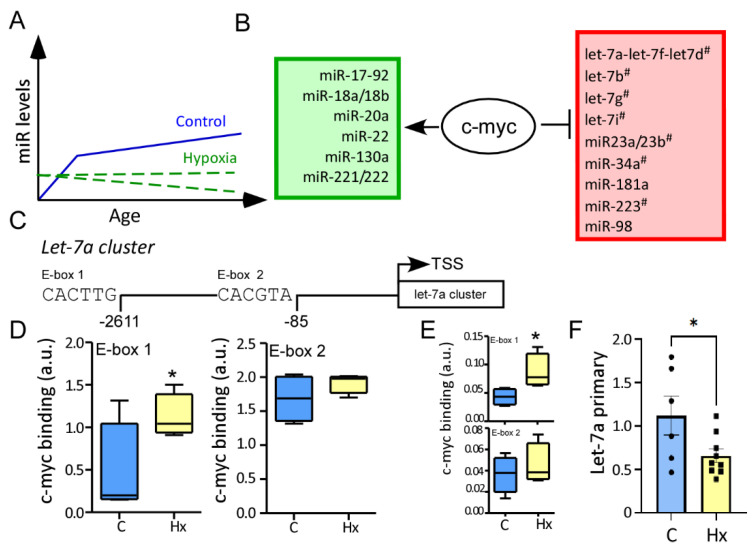
C-myc repress the expression of let-7a and miR-23a cluster. (**A**) Schematic diagram representing the main response of microRNA regulation post-hypoxia, even if the microRNA in up-regulated at one single time point, e.g., 24 h, overall the microRNA expression is stable or down regulated, in contrast these microRNAs have an increase expression between ages in the control group. (**B**) List of published microRNAs regulated by c-myc, either enhance (green) or repress (red) their expression. (**C**) Schematic showing c-myc E-Box consensus sites in the Let7a cluster promoter region in the mouse genome. (**D**) Increased c-myc occupancy of c-myc consensus sites in the let-7a promoter in the hippocampus from control and hypoxia mice 72 h post-procedure (*n* = 4/group) (**E**) Increased c-myc occupancy of c-myc consensus sites in the miR-23a promoter in the hippocampus from control and hypoxia mice 72 h post-procedure (*n* = 4/group).Note: In both promoter c-myc has a higher occupancy on the E-box 1 region and not in the E-box 2 region. (**F**). Relative expression of the primary sequence of Let-7a in hippocampus in control and hypoxia group 72 h post procedure. *n* = 6, control *n* = 9, hypoxia. Un-paired *t*-test, * *p* < 0.05.

## Data Availability

The data presented in this study are available on request from the corresponding author.

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
