# Peer review of "MicroRNA Profiling Shows a Time-Dependent Regulation within the First 2 Months Post-Birth and after Mild Neonatal Hypoxia in the Hippocampus from Mice"

_biomedicines, 2022, doi:10.3390/biomedicines10112740_

Round 1

Reviewer 1 Report

Disorders of brain development can lead to many diseases. In this manuscript, Leavy et al. generate a microRNA profile in the hippocampus and analysis their expression change after hypoxia. These microRNA identified are valuable for future investigations. There are some concerns to be addressed.

1 In Discussion, Line 343, “MiR-20a is critical for brain maturation during embryogenesis by regulating axonal growth and dendritogenesis, and its expression is time-dependent.” References are missing.

2 What is the specific method for screening the transcription factor c-myc, which need to be clarified in the methods.

3 the authors need to explain why the hypoxic model is used. In other words, The authors should discuss what developmental abnormalities may occur under hypoxic conditions.

Author Response

We would like to thanks the reviewer for the positive comments.

  1. In Discussion, Line 343, “MiR-20a is critical for brain maturation during embryogenesis by regulating axonal growth and dendritogenesis, and its expression is time-dependent.” References are missing.

Thank you to the reviewer for highlighting the omission of a reference. Reference has now been added in Line 380 at the end of the sentence.

          2 What is the specific method for screening the transcription factor c-                  myc, which need to be clarified in the methods.

The details on the methods for screening c-myc has been added to the methods, under a new section (Lines 176-182):

Identification of c-Myc binding sites in the promoter regions.

The promoter region from the 15 misregulated microRNAs over time in the hypoxia group compared to the control was examined for potential Myc binding sites. Potential C-Myc binding sites were identified by searching for Myc-binding motifs within 5000 bases upstream of the transcriptional start sites of genes of interest, using the UCSC genome browser (version: GRCm39/mm39). Primers were then designed to flank the predicted Myc binding sites.”

         3 the authors need to explain why the hypoxic model is used. In other                 words, the authors should discuss what developmental abnormalities may           occur under hypoxic conditions.

More details on the developmental abnormalities after neonatal hypoxia in our model have been added in the results section (Lines 252-254):

Previously we have shown that pups under hypoxia develop long-term neurological outcomes, including, anxiety-like behavior, hippocampal dysfunction, and a lower threshold to develop seizures [22].”

Reviewer 2 Report

Dear Authors,

Congratulations on your interesting paper. Overall, the study is novel and well-written however I have some comments:

1. Please give the exact number of animals used in the study.

2. Figures should be enlarged for greater readability.

3. What do you mean by "n=3-4/group"? is it 3 or 4 or maybe 3.5? this should be clarified.

4. Please add the power and the limitations of the study at the end of Discussion section.

Author Response

Thank you the reviewer for the kind words.

  1. Please give the exact number of animals used in the study.

The total number used on the study has been added in the first section on the methods (Line 109):

A total of 70 mice were used in this study

  1. Figures should be enlarged for greater readability.

Figures with high resolution will be provided to the editorial office to increase readability.

  1. What do you mean by "n=3-4/group"? is it 3 or 4 or maybe 3.5? this should be clarified.

N numbers per group have been clarified in the figure legends, for example, Line 357 “n=6/control; n=9/hypoxia

  1. Please add the power and the limitations of the study at the end of Discussion section.

We have added a section to identify the limitations of the study, in our conclusions we have indicated the main discoveries of current study (Lines 397-410)

The current study presents some limitations, first, we analysed a small number of microRNAs, selected by their expression on the postanal brain, we cannot discard that important microRNAs involved mainly in the maintenance of brain function, may have a higher expression at P42, and were not evaluated in the current study. Future studies should evaluate microRNA expression in elder mice to consider that microRNA expression may also be regulated from adulthood to the elderly. In the current study we analyzed male and female mice together, we observed no differences in expression between mice, however we cannot discard sex differences. Further studies should evaluate sex differences during postnatal brain development to evaluate the effect of sex in microRNA expression. Finally, we focus on the hippocampus, a vulnerable region to hypoxia, similar changes in specific microRNAs have been seen in other brain regions, e.g. entorhinal cortex [26], we can speculate that common mechanisms are activated independent of the region, however, it will be necessary to evaluate their regulation in parallel to determine if the time scale of brain maturation is similar

Reviewer 3 Report

1.     In the affiliation 2 please verify and correct the email address.

2.     In the abstract, (Line 12) substitute “normal” with the more appropriate term “physiological”.

3.     In the abstract, a long introduction of the background is followed by a method and conclusion section, but the results are missing. Please reduce the background and add one or two sentences with a brief summary of the main results. 

4.     There is no need to anticipate the results in the introduction (lines 89-99). Please focus this paragraph on the aim of the study.

5.     How where the 215 microRNAs selected?

6.     The authors start the materials and methods with the “RNA isolation and Open Array” paragraph 2.1., by describing how the RNA was extracted and processed. Reading this first paragraph it is unclear, however, where the RNA was extracted from. I’d advice the authors to start the section 2 with a paragraph introducing the subjects of the study, in this case the mice. 

7.     There’s no description of when and how the hippocampus were harvested from the mice brains. How where the hippocampus isolated? Did the authors use one hippocampus per mouse or they pulled right and left hippocampus together? How were the tissue homogenized before RNA extraction? Were the animals perfused? Those informations are important and should be included.

8.     RNA quality and quantity were measured using the Nanodrop which, sometimes, does not give a perfect reading for samples that are not pure. Have the authors considered to check the quality by running the bioanalyzer?

9.     I would suggest the authors to split paragraph 2 in two different paragraphs, one for the animals and the other one for the hypoxia procedure. 

10.  It is unclear if the study was performed on male, female (or both). In line 138, the authors state “Male and female pups”. It’d be useful to know if both genders were investigated and/or if any gender-specific microRNA expression was observed.

11.  It is unclear when the ambulation score test was performed. In the Figure legend 3 the authors state “Pups subjected to hypoxia have a lower 296 ambulation score 24 hr post-hypoxia compared to the control pups”, while in the paragraph 3.3. “ambulation test were performed 72 hours post procedure”. 

Author Response

We would like to thank you the reviewer for the attention to detail.

1. In the affiliation 2 please verify and correct the email address.

We would like to apologize for this mistake, email address has been removed.

2. In the abstract, (Line 12) substitute “normal” with the more appropriate term “physiological”.

In the abstract “normal” has been changed to “physiological”

3.In the abstract, a long introduction of the background is followed by a method and                           conclusion section, but the results are missing. Please reduce the background and add one or two sentences with a brief summary of the main results.

We have reduced the long introduction, and we have added a summary of the results (lines 22-25):

“We observed that microRNAs can be divided in four groups based on their temporal expression. Further after an early-life insult, hypoxia at P7, 15 microRNAs showed a misregulation over time, including Let7a”

4. There is no need to anticipate the results in the introduction (lines 89-99). Please focus this paragraph on the aim of the study.

We have added the information of the aim of the study (lines 91-93):

We aim to evaluate how microRNA expression changes over time in the hippocampus and how an early life insult, hypoxia at Postnatal day 7 may result in the misregulation of microRNAs and contribute to the neurological outcomes”

Also, we have kept a small section on how we approached their aims, to make it clear to not experts in their field.

5.How where the 215 microRNAs selected?

The 215 microRNAs were selected based on the expression on the postnatal brain, as this is a limitation of the study, this has been added to the discussion section (lines 397-402):

“... we analysed a small number of microRNAs, selected by their expression on the postanal brain, we cannot discard that important microRNAs involved mainly in the maintenance of brain function, may have a higher expression at P42, and were not evaluated in the current study. Future studies should evaluate microRNA expression in elder mice to consider that microRNA expression may also be regulated from adulthood to the elderly

6. The authors start the materials and methods with the “RNA isolation and Open Array” paragraph 2.1., by describing how the RNA was extracted and processed. Reading this first paragraph it is unclear, however, where the RNA was extracted from. I’d advice the authors to start the section 2 with a paragraph introducing the subjects of the study, in this case the mice.

The methods section has been organized following the reviewer’s advice, now, we start with the mouse model and continue with the Open Array.

7.There’s no description of when and how the hippocampus were harvested from the mice brains. How where the hippocampus isolated? Did the authors use one hippocampus per mouse or they pulled right and left hippocampus together? How were the tissue homogenized before RNA extraction? Were the animals perfused? Those informations are important and should be included.

Information in how the hippocampi were harvested has been added to the manuscript (lines 127-128):

For each time point and experimental condition, mice were perfused with cold PBS and both hippocampi were isolated and pooled together for RNA extraction

8. RNA quality and quantity were measured using the Nanodrop which, sometimes, does not give a perfect reading for samples that are not pure. Have the authors considered to check the quality by running the bioanalyzer?

We agreed with the reviewer that using a bioanalyzer will provide more accurate information about the quality of the samples, and nanodrop cannot provide information on the degradation of the RNAs. However, all our samples were processed in parallel, and we don’t expect differences on their quality. Differences in the quality may be critical for changes in the expression between groups, however, we don’t expect this in these experiments, as all samples were prepared in parallel and kept in dry-ice or -80C.

9.I would suggest the authors to split paragraph 2 in two different paragraphs, one for the animals and the other one for the hypoxia procedure.

Animal procedures have been separated in two different paragraphs.

10. It is unclear if the study was performed on male, female (or both). In line 138, the authors state “Male and female pups”. It’d be useful to know if both genders were investigated and/or if any gender-specific microRNA expression was observed.

Study was performed in males and females and both sexes were included in our sample, however, we don’t have enough n numbers to evaluate sex differences. This is a limitation of the current study and it has been added in the discussion section (lines 402-406):

“In the current study we analyzed male and female mice together, we observed no differences in expression between mice, however we cannot discard sex differences. Further studies should evaluate sex differences during postnatal brain development to evaluate the effect of sex in microRNA expression.”

11.It is unclear when the ambulation score test was performed. In the Figure legend 3 the authors state “Pups subjected to hypoxia have a lower 296 ambulation score 24 hr post-hypoxia compared to the control pups”, while in the paragraph 3.3. “ambulation test were performed 72 hours post procedure”.

Ambulation test was performed 72 hours post-hypoxia, this has been corrected on the figure legend